# Prevalence of zolpidem use in France halved after secure prescription pads implementation in 2017: A SNDS database nested cohort study

**Pascal Caillet** [1] *, **Morgane Rousselet** [1,2], **Marie Gerardin** [1], **Pascale Jolliet** [1,2], **Caroline Victorri-Vigneau** [1,2]

1 Department of Clinical Pharmacology, Addictovigilance, University Hospital of Nantes, Nantes, France,
2 SPHERE U1246 Unit, University of Nantes, University of Tours, INSERM, Nantes, France

* pascal.caillet@chu-nantes.fr

## Abstract

Our objective was to quantify the impact on the use of zolpidem of the obligation implemented in France in 2017 to use secure prescription pads to prescribe it. We conducted a cohort study within the French SNDS healthcare database. Patients aged over 18 years of age were considered for inclusion. The number of prevalent users and incident episodes of zolpidem use were compared before the change in law (July 1, 2016 to January 1, 2017) and after (July 1, 2017 to January 1, 2018). A prevalent user was a patient who has been reimbursed for zolpidem at least once. An incident episode of zolpidem use was defined by a first administration of zolpidem without any prior administration within the previous six months. Regarding prevalence of zolpidem users, we observed a decrease from 2.79% (CI95%:2.75–2.83) to 1.48% (1.44–1.51), with a number of patients who stopped taking it after the change in law being approximately 4.3 times higher than the number of patients who started. We observed a negative association between the post-law change period (OR = 0.52 (0.51–0.53)) and the probability of receiving zolpidem, adjusting for sex, aging, low income and chronic disease. We observed a decrease from 183 treatment episodes per 100,000 insured months on average to 79 episodes per 100,000 insured months, with an incidence rate ratio (IRR) equal to 0.43 (0.38–0.49). The use of secure prescription pads seems to have reduced the exposure of the French population to zolpidem.

## Introduction

Zolpidem, a hypnotic drug marketed in France since 1987 (Stilnox®) is an imidazopyridine which binds to the benzodiazepine binding site on the GABA-A receptors. It is highly selective for the α1 subtype receptor and initial clinical trials reported no evidence of abuse or dependence [1,2]. As a consequence, zolpidem has been marketed as a safer drug than benzodiazepines, particularly regarding dependence issues.

The first case report describing dependence on zolpidem appeared in the literature in 1993 [3]. The French health authorities (National Agency for Medicines and Health Products Safety;

**Data Availability Statement:** Data are available from the French Caisse Nationale de l'Assurance Maladie (CNAM) for researchers who meet the

criteria for access to confidential data. These are edicted by the French national commission governing the application of data privacy laws (the Commission Nationale Informatique et Libertés). Consequently, data are available after obtaining legal authorization from the French Institut des Données de Santé (at https://www.indsante.fr/) and from the French national commission governing the data privacy laws (Commission Nationale Informatique et Liberté; CNIL, https://www.cnil.fr/).

**Funding:** This work was funded by the French ANSM institution (AAP-2017-027) and the University Hospital of Nantes. The funders had no role in study design, data collection and analysis, decision to publish, or preparation of the manuscript.

**Competing interests:** The authors have declared that no competing interests exist.

ANSM) launched an official study in 2002 to evaluate the dependence potential of zolpidem. This study was conducted by Nantes Centre for Evaluation and Information on Pharmacodependence (CEIP). The 3 main CEIP missions are to collect data and assess the dependence potential of the identified psychoactive drugs, to provide information on the risk of abuse or dependence on psychoactive substances, and to conduct research. The results highlighted the high dependence potential of zolpidem and identified 2 distinct populations among dependent patients. The first type seeks paradoxical stimulant effects by taking high doses during the day. The second type includes dependent patients who have been treated for insomnia, but who, given the short half-life and tolerance, have increased their doses [4]. As a result, the summary of product characteristics of zolpidem was comprehensively modified in 2004, with the inclusion of the sentence: "A pharmacodependence may materialize even at therapeutic doses, and/ or for subjects who do not show an individualized risk factor" [1]. In June 2012, an update of these data found the same results, but suggested escalation in number and gravity of cases, with particularly high dose intake. Moreover, zolpidem has been more and more involved in forged prescription sheets, being present in 29% of cases of forgery reported in the French survey of Suspect Prescriptions Possibly Indicating Abuse (OSIAP for "*Ordonnances Suspectes Indicateur d'Abus Possible*" in French) over the 2009 to 2012 period [5].

In light of these results, the prescription of zolpidem on secure prescription pads has been put forward by the French National Commission on Narcotics and Psychotropic Substances. Secured prescription pads contain specific prescription sheets that are protected against forgery and that must be used by physicians when prescribing narcotics. On the 11th of January 2017, the ANSM decreed that as from April the 10th 2017, the prescription of zolpidem was to be done with such secured prescription pads in order to limit the risk of abuse and misuse. Moreover, such prescription must now display the number of therapeutic units, quantity and dosage written in full and for a maximum duration of 28 days. The objective of the ZORRO study (ZOlpidem and the Reinforcement of the Regulation of prescription Orders) was to evaluate the overall impact of the obligation to use secure prescription pads for zolpidem. The present work aimed at evaluating the impact of this measure on the number of zolpidem consumers.

## Materials and methods

### Experimental design

This study was an observational prospective study using the French healthcare data system (SNDS)[6], conducted by the Nantes University Hospital Addictovigilance department. The study was founded by a grant from the ANSM (AAP-2017-027) and was monitored by a pluridisciplinary steering committee with pharmacologists, psychiatrists specialized in addiction, pharmacoepidemiologist and general practiners (GP).

### Participants and outcomes

The study sample included all patients in the Generalist Sample of Beneficiaries (EGB). EGB is a 1/97th representative sampling of the SNDS. The SNDS links several existing databases: the nationwide claims database of the French National Healthcare system (*Système National d'Informations Interrégimes de l'Assurance Maladie; SNIIRAM)*; the national hospital database (Programme de Médicalisation des Systèmes d'Information; PMSI) and the national death registry (Centre d'épidémiologie sur les causes médicales de Décès; CepiDC). The SNDS covers more than 98% of the French population (66 million people) from birth (or immigration) to death (or emigration), even in case of change in occupation or retirement. Data are individual and anonymous. The SNDS contains a longitudinal record of health encounters, hospital

diagnoses and drugs deliveries relative to outpatient medical care claims, including all reimbursed drugs, information from hospital discharge summaries, and date of death.

The primary outcome was to quantify the number of zolpidem consumers and the rate of zolpidem treatment initiation before and after the implementation of the secured prescription. The secondary outcome was the identification of factors associated with zolpidem consumption after the change in law.

## Data collection

The target population of our research was consumers of zolpidem included in the EGB aged over 18 years old. A prevalent user was defined as a patient being reimbursed at least once for zolpidem and an incident episode of zolpidem use was defined as a patient receiving a first delivery of zolpidem without any prior deliveries over the preceding 6 months.

Two time periods were used: *(i)* period 1 from July 1, 2016 to January 1, 2017, reflecting zolpidem consumption before the change in law *(ii)* period 2 from July 1, 2017 to January 1, 2018 reflecting the steady state of zolpidem consumption after the change in law.

The variables gathered in the database included age at inclusion, sex, information on active CMUc (for "Couverture médicale universelle complémentaire", which indicates low-income status granting full reimbursement of health expenditures), and the presence of a chronic disease *via* registration in the list of chronic diseases scheme beneficiaries (ALD for "Affections de Longue Durée").

## Ethical issues

This study was approved by the French Committee of Protection of Persons (*Comité de Protection des Personnes*, CPP, approval reference 2018-A01070-35) and the National Data Protection and Freedoms Committee (approval reference 918201). The study was registered on www.clinicaltrial. gov under the reference NCT03584542. The study protocol is available as an open publication [7].

## Statistical analysis

Prevalences were described with their point estimates and 95% accuracy intervals. The comparison of prevalences between both periods was performed with a McNemar test. Let's consider probabilities of respectively stopping treatment in period 2 when you are a consumer in period 1($p_{stop}$) and starting treatment in period 2 when you did not consume in period 1 ($p_{start}$). The null hypothesis of the test is $p_{stop}$ equal $p_{start}$. The alpha risk was set at 5% and the statistical significance threshold set at 5%. This analysis was done on the dataset containing only patients present in the EGB database over the 2 periods. A sensitivity analysis was conducted by recoding patients missing in a period as non-consumers in the period considered. Incidences were first described by constructing rates of treatment initiations per 100,000 insured per month (patient-month). The description of incidences trends was then conducted through the construction of incidence rate ratios (IRRs) adjusted for the time period with a negative binomial model.

In order to study effects of potential confounders of the effect of change in law, a logistic regression was conducted. The variables to be included in the model were defined *a priori* and included period (before vs after change in law), age in 2016, sex, being under a CMUc scheme or an ALD scheme. Age was introduced into the model in class intervals because the hypothesis of a constant effect of aging on the probability of receiving zolpidem regardless of the age stratum in which patients are located was unlikely to be respected for this variable. The adequacy of the model to the data was checked graphically. The dependencies between observations in a same patient over the two time periods were handled by the use of Generalized

**Table 1. Characteristics of the population.**

| Period | July 1, 2016 to January 1, 2017 | July 1, 2017 to January 1, 2018 |
|---|---|---|
| Number of patients | 545,478 | 552,935 |
| Age at inclusion (mean, SD) | 50.0 (19.1) | 50.1 (19.2) |
| Male | 48.4 | 48.5 |
| ALD Status | 20.0 | 20.3 |
| CMUc Status | 4.9 | 4.8 |
| Zolpidem consumers (N; %; (IC95%)) | 15,222; 2.79; (2.75–2.83) | 8,165; 1.48; (1.44–1.51) |

Estimating Equations (GEE), with the use of a robust variance estimator in order to facilitate convergence and improve the accuracy of confidence intervals. All analyses were performed with SAS 9.4 (SAS Institute, North Carolina, USA).

## Results

### Study participant description

The characteristics of the population present in the database each year are described in Table 1. Albeit an overall increase in the population size is observed, the characteristics remained similar between the two periods, with a population showing a sex ratio of 0.96, aged 41 years on average at inclusion, one sixth benefiting from the ALD scheme and 6% benefiting from a CMUc scheme.

### Primary outcome

**Prevalences.** Regarding the variation in prevalence of zolpidem consumers, we observed a decrease from 2.79% to 1.48% (Table 1). This variation in prevalence seemed to be mainly explained by a high number of treatment drops, with a number of patients stopping consumption between the 2 periods (n = 9018) about 4.3 times higher than the number of patients starting during period 2 (n = 2107) (Table 2). The sensitivity analysis showed very similar results (Table in S1 Table).

**Incidences.** Regarding variation in incidence of new zolpidem treatments we observed a 57% decrease in the incidence of new treatment episodes between the period 1 and the period 2 (from 183 to 79 treatment initiations per 100,000 insured months respectively), with an Incidence Rate Ratio (IRR) of 0.43 (0.38–0.49) (Fig 1). The announcement of the measure was not associated with a statistically significant decrease in the number of treatment initiations (IRR = 0.92 (0.79–1.08) when comparing the period 1 vs period from January 2017 to April 2017). The decrease became statistically significant only when the measure application became mandatory (IRR = 0.40 (0.34–0.47) when comparing period 1 vs period from April 2017 to July 2017).

**Table 2. Mc Nemar test regarding changes in prevalence consumptions.**

| Consumer in period 1 | Consumer in period 2 | | |
|---|---|---|---|
| | No | Yes | Total |
| No | 520,397 (96.81%) | 2,107 (0.39%) | 522,504 (97.21%) |
| Yes | 9,018 (1.68%) | 6,004 (1.12%) | 15,022 (2.79%) |
| Total | 529,415 (98.49%) | 8,111 (1.51%) | 537,526 (100.00%) |

Mc Nemar's Test: Chi-Square 4293.2064; DF 1, p-value <0.0001

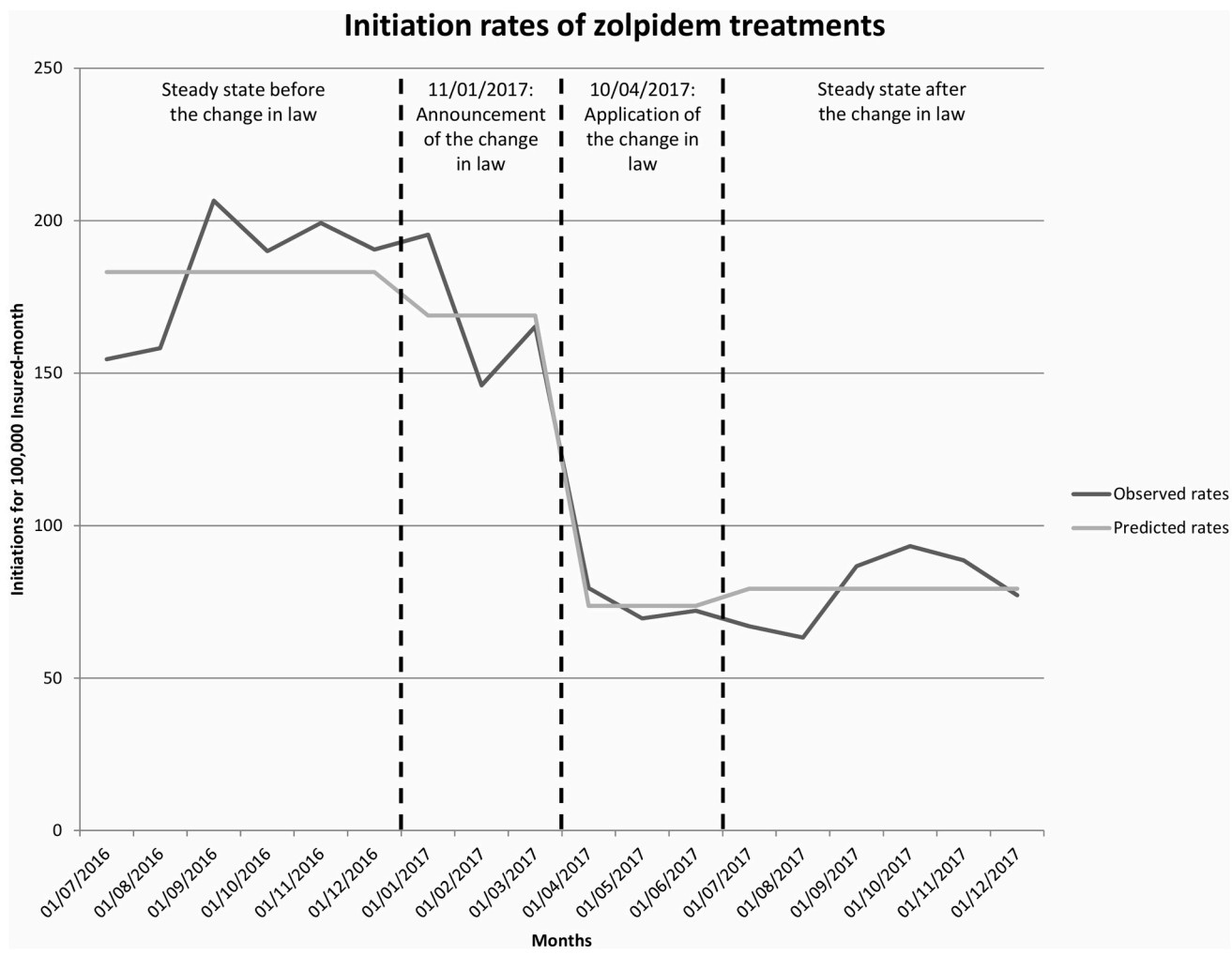

**Fig 1. Rates of zolpidem initiations between July 2016 and January 2018.**

## Secondary outcome

**Evaluation of consumption-associated factors.** The characteristics of the population included in the model and the results are described in Table 3. Regarding age, values of odds ratios (OR) ranged from 2.05 (1.86–2.27) for 30-40y old vs 18-30y old comparison to 8.86 (8.09–9.69) for ≥70y old vs 18-30y old comparison. There was also a positive association between being female, benefiting from ALD or CMUc, and the likelihood of receiving zolpidem. On the contrary, a strong negative association was observed between the post change-in-law period (OR = 0.52 (0.51–0.53)) and the likelihood of receiving zolpidem.

## Discussion

We found that the change in law that occurred in the beginning of 2017 regarding zolpidem prescription was concomitant with the halving of the prevalence of zolpidem users. This association was independent of age, sex, from suffering from a chronic disease granting an ALD, and from being precarious enough to benefit from the CMUc scheme. We also observed that the incidence of new zolpidem treatment episodes halved after the promulgation of the law.

**Table 3. Results of the multiple logistic regression on patients present in both periods and aged 18 or over in 2016 (N = 535,605).** Being a consumer is predicted.

| Variable | 2016 (Size, %) | 2017 (Size, %) | Adjusted Odds Ratio (CI95%) |
|---|---|---|---|
| Age | | | |
| 18-<30 | 90,052 (16.8) | | Reference |
| 30-<40 | 90,210 (16.8) | | 2.05 (1.86–2.27) |
| 40-<50 | 94,292 (17.6) | | 3.75 (3.42–4.12) |
| 50-<60 | 90,632 (16.9) | | 5.74 (5.25–6.28) |
| 60-<70 | 80,537 (15.0) | | 7.20 (6.58–7.88) |
| 70+ | 89,882 (16.8) | | 8.86 (8.09–9.69) |
| Study period relative to the change in law | | | |
| Before | | | Reference |
| After | | | 0.52 (0.51–0.53) |
| Sex (Female vs Male) | | | |
| Male | 259,165 (48.4) | | Reference |
| Female | 276,440 (51.6) | | 1.67 (1.61–1.72) |
| Chronic disease (ALD) | | | |
| No | 430,649 (80.4) | 423,929 (79.2) | Reference |
| Yes | 104,956 (19.6) | 111,676 (20.8) | 1.98 (1.91–2.05) |
| Low income (CMUc) | | | |
| No | 509,072 (95.0) | 510,700 (95.4) | Reference |
| Yes | 26,533 (5.0) | 24,905 (4.6) | 2.02 (1.90–2.14) |

The main strength of this study is that we have provided recent population-level data across a representative sample of the exhaustive national reimbursement database. The validity and usefulness of this database in pharmaco-epidemiological studies have been previously studied and validated, and the analysis of large medico-administrative databases offers the advantages of completeness of patients and unbiased study sample [6]. The main limitation is the total absence of clinical information, e.g. concerning the effects searched for and felt by patients, the modification in routes of administration, the tolerance and the withdrawal. We have no information either about therapeutic strategies used to reduce zolpidem use. As we rely on a medico-administrative reimbursement database, we have no information on illegal drug use or purchase of zolpidem or other therapies. A complementary in-field study is currently ongoing to document these points [7].

These initial results showed that the enforcement of the use of secured prescription sheets had an effect on the population's exposure to zolpidem. Since there are cues that prescription drug abuse varies with the availability of the involved medication [8], our results suggest that the change in law could be effective in limiting the entry of new users into abuse or dependence. These results are concordant with observations of primary care pharmacists, albeit in a more restricted area [9]. Among the 235 responding pharmacies in the French Rhône area, zolpidem dispensation decreased by an average of 41.8%. On the same time windows, zopiclone dispensations for the 3.75 and 7.5mg dosages increased by 18.4 and 16.7% respectively. Our results were also concordant with another study using the French National Health Insurance database. One year after the information of the ANSM about the measure, the prevalence reimbursement for zolpidem among individuals who had at least one hypnotic drug reimbursement decreased from 26.0% to 18.4%, while the prevalence of reimbursement for zopiclone increased from 18.0% to 28.3% [10].

This impact has far exceeded its initial purpose, *i.e.* reducing the population of zolpidem misusers. From the practitioners' standpoint, this could be at least partly explained by the manifold ways in which they appropriated this reform.

First, the communication by the health authorities regarding what motivated the measure provided important information regarding risks associated to zolpidem treatment [11,12]. Following this information, a readjustment by the practitioner of the treatment's benefit/risk balance at the time of its first prescription or renewal could have occurred and could also have triggered open discussions regarding patients' chronic consumptions. For patients considering their consumption as having no negative effects, the fact that a regulatory change occurred for misuse control motives could have been a useful tool for physicians in their discussion to prove the contrary. They may have been to be able to make patients initially reluctant to stop their zolpidem consumption more aware of their problematic use and to agree a try to gradually reduce doses. All this could have led to an increased detection and management of substance use disorders, including withdrawal under medical supervision. This could at least partially explain why the measure has had a far greater impact than initially expected (the proportion of problematic zolpidem user was estimated to 1% of chronic consumers in a previous study)[13].

Second, this reform has added constraints to the prescription process of a drug that is often used in clinical practice. The implementation of a less administratively burdensome alternative (e.g. zopiclone) is to be expected, regardless of the treated patients' profile, as general practice consultations last an average of 15 minutes on average [14] and an extension of this duration for purely administrative reasons may not be acceptable from a practitioner's point of view. This kind of optimization of consultation time seems all the more plausible as physicians are often over-solicited.

Moreover, adding constraints on the prescription process seems to be more impactful in decreasing drug exposure than simply informing physicians regarding a specific risk [15] (e.g. restricting the target population by adding contraindication vs increasing awareness to risks by addition of safety precautions in the summary of the product of a drug) [16]. A more appropriate multidimensional approach could have a specific impact on target subgroups, without spilling over into other populations for which the prescription of the drug could have been useful.

From the patients' standpoint, recreational user and self-medicating user profiles may have had different adaptations in the face of the regulatory tightening. As an example, recreational users may have switched to other drugs worse than zolpidem, which may have turned the measure counter-productive in this specific subpopulation. The real impact on patients suffering from zolpidem addiction is still largely unknown and SNDS data do not allow this point to be accurately documented. However, the principles of management of dependence with 'z-drugs' such as zolpidem and zopiclone are the same as the management of benzodiazepine dependence [17]. An interesting point would be to monitor the use of other alternative therapies (e.g. zopiclone, benzodiazepines) to better understand the decrease in the use of zolpidem after the measure. Another lead could be to monitor changes in inadequate behavior, such as doctor shopping and use of excessive doses, which already proved to be useful in studying drug misuse using the SNDS database [13,18]. These investigations are currently conducted. All these point are crucial in order to evaluate if the change-in-law had a beneficial effect, both in terms of individual health and public health.

Regarding the associations between zolpidem consumption and sex, older age, low income and presence of comorbidities, these are consistent with what has already been observed in 2015 on data issued from the same database, but analyzed by another team [19]. These associations are also observed in other settings, which suggest that these are not specific to the French

population [20,21]. It should be noted that zolpidem is the most frequently prescribed hypnotic drug in the elderly [19], for whom the risk of dependence is high [22,23]. Studies investigating the evolution of consumption in this specific population are needed.

Secure prescription pads have already been previously implemented in France for other drugs such as flunitrazepam in 2001 and clonazepam in 2011. These regulatory measures were first implemented to reduce misuse, but ultimately had broader implications: reducing the global exposure to the drug in question and changing all other hypnotic drugs prescriptions (e.g. switching to other drugs) [19,24].

Misuse, abuse or dependence on zolpidem is a worldwide problem [25–27]. In France, regulatory changes such as the addition of prescription constraints seem to have a higher impact on prescriptions than the information for physicians. The secured prescription pads used for zolpidem prescription could be replicated in other countries who have to deal with this public health issue. Patients information may be another effective way to reduce the use of zolpidem and especially in countries where drug advertising is allowed (e.g. in United States). Indeed, the expected impact can be considerably modified by the media coverage of the public health decision being implemented [28–30].

## Conclusions

The enforcement of the use of secured prescription sheets seemed to have decreased French population's exposure to zolpidem. Additional studies on the transfer of consumption to other drugs and the evolution of the type of consumption (chronicity and problematic use) are currently being conducted using the SNDS. Complementary to the use of medico-administrative data, a field study among problematic consumers is still in progress to clinically assess the impacts of the measure. All these components put together form an approach which is the most original to date regarding the evaluation of a measure aiming at limiting hypnotic abuse at a whole country level.

## Supporting information

**S1 Table. Mc Nemar test regarding changes in prevalence consumptions (sensitivity analysis).**
(DOC)

## Acknowledgments

We wish to thank Pierre Loué and Marion Istvan for their contribution to this work.

## Author Contributions

**Conceptualization:** Pascal Caillet, Morgane Rousselet, Marie Gerardin, Caroline Victorri-Vigneau.

**Data curation:** Pascal Caillet.

**Formal analysis:** Pascal Caillet.

**Funding acquisition:** Pascal Caillet, Morgane Rousselet, Pascale Jolliet, Caroline Victorri-Vigneau.

**Investigation:** Pascal Caillet, Morgane Rousselet, Marie Gerardin, Caroline Victorri-Vigneau.

**Methodology:** Pascal Caillet, Morgane Rousselet, Caroline Victorri-Vigneau.

**Project administration:** Pascal Caillet, Morgane Rousselet, Marie Gerardin, Caroline Victorri-Vigneau.

**Resources:** Pascal Caillet, Pascale Jolliet, Caroline Victorri-Vigneau.

**Software:** Pascal Caillet.

**Supervision:** Pascal Caillet, Marie Gerardin, Pascale Jolliet, Caroline Victorri-Vigneau.

**Validation:** Pascal Caillet, Caroline Victorri-Vigneau.

**Visualization:** Pascal Caillet.

**Writing – original draft:** Pascal Caillet, Caroline Victorri-Vigneau.

**Writing – review & editing:** Pascal Caillet, Caroline Victorri-Vigneau.

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
