## [Decision Letter · Decision Letter 0]

5 Nov 2019

PONE-D-19-22052

Prevalence of zolpidem use in France halved after secure prescription pads implementation in 2017: a SNDS database nested cohort study.

PLOS ONE

Dear Dr Caillet,

Thank you for submitting your manuscript to PLOS ONE. After careful consideration, we feel that it has merit but does not fully meet PLOS ONE’s publication criteria as it currently stands. Therefore, we invite you to submit a revised version of the manuscript that addresses the points raised during the review process.

Both reviewers liked the manuscript and asked to consider modifying some terms, discuss other possible reasons for the decrease in the use of Zolpidem (illegal use, substitution to a over the counter Zopiclone and , the different experience in the US.

Please, address all issues raised and return a revised version for further analysis.  

We would appreciate receiving your revised manuscript by Dec 20 2019 11:59PM. To enhance the reproducibility of your results, we recommend that if applicable you deposit your laboratory protocols in protocols.io, where a protocol can be assigned its own identifier (DOI) such that it can be cited independently in the future. For instructions see: http://journals.plos.org/plosone/s/submission-guidelines#loc-laboratory-protocols

We look forward to receiving your revised manuscript.

Kind regards,

Cesario Bianchi

Academic Editor

PLOS ONE

Journal Requirements:

3. Thank you for including your ethics statement:

This study was approved by the local ethic board and the French National Data Protection and Freedoms Committee (number 918201). The study was registered on www.clinicaltrial.gov under the number NCT03584542. The study protocol is available as an open publication [8].

Additional Editor Comments:

Dear Dr. Caillet,

Your manuscript was reviewed by 2 experts that found it interesting. However, they are asking for several clarifications and the inclusion of additional references.

In particular, both reviewers are concerned about the possibility that the decrease in Zolpedem Use be related to change to other drugs like diazepam (reviewers 1 and 2 and Zopiclone (reviewer 2) that is purcahsed over the counter in some European countries.

Is there an increase in its illegal use? Are there age-groups differences?, etc...

Reviewer 1 is also asking to make a comparison between US and France because the US experience seems to be different. The reviewer has many concerns about some terms use like addition vs dependence.

Both reviewers asked to include some references relevant to the topic.

Please, carefully address every issue raised and make changes if find necessary, in your revised version.

Thank you ,

Cesario Bianchi

Reviewers' comments:

Reviewer's Responses to Questions

**Comments to the Author**

1. Is the manuscript technically sound, and do the data support the conclusions?

Reviewer #1: Yes

Reviewer #2: Partly

2. Has the statistical analysis been performed appropriately and rigorously? 

Reviewer #1: I Don't Know

Reviewer #2: N/A

3. Have the authors made all data underlying the findings in their manuscript fully available?

Reviewer #1: Yes

Reviewer #2: Yes

4. Is the manuscript presented in an intelligible fashion and written in standard English?

Reviewer #1: Yes

Reviewer #2: Yes

5. Review Comments to the Author

Reviewer #1: Overall a nice paper demonstrating that a French government intervention had its intended effect of decreasing prescriptions of the drug zolpidem due to concerns about dependence. Contrasts with the US experience related to Drug Safety Communications and zolpidem prescribing.

Abstract

Lines 16-17: Not really a comment limited to the Abstract, but why only 6 months pre- and post-?

Introduction

Line 46: Would not use the term "addict patients," that term has fallen out of favor in the US and elsewhere. Suggest sticking to the term dependent as it was used earlier in the same sentence

Lines 48-49: Should read "increased their doses considerably" -- but please clarify what is meant by considerably. Do you mean 2.5 mg to 10 mg? If so I'm not sure that's a considerable increase since it's within the recommended dose range and patients are generally titrated up to achieve therapeutic effect

Line 57: Please explain what "secure prescription pads" are. We don't have them in the United States, but this is an interesting concept and may have promise in the US depending on what they are and how they are utilized in France. Without a clear definition, it's hard to determine the broader applicability of these findings.

Materials and Methods

Lines 80-82: A lot of acronyms in the Methods, some undefined (OSIAP, SNIIRAM). Can you write out the ones that don't appear as frequently in the manuscript (as you have with PMSI and CepiDC)?

Lines 104-106: Date ranges don't make sense here. Please clarify. The second one goes (backwards?) from July 2017 to January 2017?

Primary Outcome

Line 167: The use of the term "significant" here is confusing -- unless referring to "statistical significance" I would change this term to something else. I also think there is more to be presented here in terms of the sensitivity analysis, instead of just "no significant changes."

Lines 174-178: There may be a more concise way of saying all this. Both sentences seems to be pointing to the same data points, right? Decreasing from 183 to 79 is the 57% decrease? If that is not the case, more clarification is necessary.

Discussion

Line 220: Would again avoid using the term "addiction". Dependence is more appropriate. Though I would be careful with use of the term dependence as well. How is it defined, and can your data really show that "dependence" among individual users decreased in this large, aggregate data set?

Lines 237-238: "may have successfully turned patient initially reluctant" (phrasing here doesn't read correctly and should be amended)

Lines 251-252: "in a current context where the number of active practitioners in France is insufficient to meet the primary care needs." (this phrase, though perhaps true, is not relevant to this manuscript and should be deleted)

Lines 253-254: This is an important distinction. It is notable that in the US, a Drug Safety Communication from the FDA did not result in statistically significant changes in prescribing (Kesselheim AS, Donneyong M, Dal Pan GJ, Zhou EH, Avorn J, Schneeweiss S, Seeger JD. Changes in prescribing and healthcare resource utilization after FDA Drug Safety Communications involving zolpidem-containing medications. Pharmacoepidemiol Drug Saf. 2017 Jun;26(6):712-721.) I would ask that the authors cite this paper and discuss the differences between France and the US that may have made secure prescription pads successful in France but DSCs unsuccessful in the US.

Lines 259-260: Please also cite the US paper looking at news media and zolpidem prescribing changes: Woloshin S, Schwartz LM, Dejene S, Rausch P, Dal Pan GJ, Zhou EH, Kesselheim AS. Media Coverage of FDA Drug Safety Communications about Zolpidem: A Quantitative and Qualitative Analysis. J Health Commun. 2017 May;22(5):365-372.

Lines 269-271: "An interesting point would be to monitor the use of long half-life benzodiazepines (e.g diazepam), which could give clues regarding frequency of tapering drug use after the measure" (I'm not sure what point is being made here; it should either be clarified further or deleted)

Finally, I'd like the authors to take some time in the discussion to discuss the broader global implications of such a finding. Have secure prescription pads been used for other drugs in France? What was the outcome? Could this be replicated in a place like the United States, where health systems are far more fragmented? A succinct policy take-home would be nice. As the majority of cites are for French studies, a review of the US (and other EU) literature is warranted.

Conclusion

Lines 284-287: Sentence does not read coherently as written and needs to be rephrased.

Line 285: Would not use the phrase "abuser profiles"

Figure 1: Would use the month in the x-axis, not the date as shown. For example, July 2016, not 01/07/2016.

Figure 1: Please label the dashed lines more appropriately. I can't tell what 11/01/2017 and 10/04/2017 refer to here (and again, the US/EU conventions for dates leave me confused as to when these events happened.

Reviewer #2: This study aimed to analyse the effect of a change in French law that categorised Zolpidem to a higher level of prescription control in January 2017. Prior to the law change, Zolpidem use had been associated with tolerance and dependence mechanisms when administered in both a clinical and recreational setting. The law was changed to permit Zolpidem use only with Secure Prescription pads.

Zolpidem is a non-benzodiazepine drug that binds to GABA-A receptors at the same site as traditional benzodiazepines. It acts as an allosteric modulator to potentiate the action of the natural agonist GABA and increase the channel opening frequency allowing chloride ions to enter the cell and hyperpolarise the membrane. The effects of this are varied including, sedation, anxiolysis, anaesthesia and anticonvulsant. Zolpidem being different structurally to benzodiazepines has preference for the alpha-1 subtype of the GABA-A receptor and thus causes a sedation effect without the other benzodiazepine associated effects. Interestingly Zolpidem has been associated in some cases with feelings of euphoria and hallucinations upon administration, an effect distinct from the GABA-A action, when co-administered with BZ, but when BZ was withdrawn, the euphoria persisted [1]. Perhaps Zolpidem may show secondary targets.

This informatics study selects data from the General Sample of Beneficaries which is further connected to 3 other databases that banks patient information from over 98% of the French population. Participants of over 18 years of age were selected and data was analysed for ’prevalent users’ (users that had been reimbursed for zolpidem at least once before) and ‘incident users’ (users that has started afresh Zolpidem use). These parameters were compared with factors such as age, sex, health/disease state.

The authors found that the change in law showed a negative correlation with Zolpidem use both in prevalent use and incident use.

Criticisms:

- The prevalent user data could not be analysed for separate age groups? Dependence in the elderly population may be higher due to reduced half-life of Zolpidem. There are many reports of Zolpidem toxicity in the elderly [1][2][3].

- The authors mention themselves that the study is limited by the lack of information on how use has changed in correlation to use of alternative therapies, which may or may not be more dangerous (such as opioid or benzodiazepine use). Is it possible for the authors to access participant records to check for increase in prescriptions of these to find if there is correlation.

- The study is also limited in its lack of information regarding drug use/purchase (of both zolpidem or other therapies) outside of the awareness of the medical practitioners prescribing. Has illegal use/purchase of zolpidem increased in this time period?

- The authors could collect data on users over the period regarding what therapeutic strategies were used to reduce Zolpidem use, perhaps finding correlation with replacement therapies both pharmacological and psychological.

- Zopiclone is an alternative to Zolpidem and in some European countries is available without prescription. How can we be certain that users are not switching to Zopiclone?

1. Heydari M, Isfeedvajani MS. Zolpidem dependence, abuse and withdrawal: A case report. J Res Med Sci. 2013;18(11):1006–1007.

2. Pourshams M, Malakouti SK. Zolpidem abuse and dependency in an elderly patient with major depressive disorder: a case report. Daru. 2014;22(1):54. Published 2014 Jul 10. doi:10.1186/2008-2231-22-54

3. Styliani, S. , Ioannis, D. , Jannis, N. , Apostolos, I. and Georgios, K. (2009), Zolpidem Dependence in a Geriatric Patient: A Case Report. Journal of the American Geriatrics Society, 57: 1962-1963.

6. PLOS authors have the option to publish the peer review history of their article (what does this mean?). If published, this will include your full peer review and any attached files.

Reviewer #1: No

Reviewer #2: No

---

## [Author Response · Author response to Decision Letter 0]

28 Dec 2019

PONE-D-19-22052

 Prevalence of zolpidem use in France halved after secure prescription pads implementation in 2017: a SNDS database nested cohort study.

 PLOS ONE

 Dear Dr Caillet,

Thank you for submitting your manuscript to PLOS ONE. After careful consideration, we feel that it has merit but does not fully meet PLOS ONE’s publication criteria as it currently stands. Therefore, we invite you to submit a revised version of the manuscript that addresses the points raised during the review process.

Both reviewers liked the manuscript and asked to consider modifying some terms, discuss other possible reasons for the decrease in the use of Zolpidem (illegal use, substitution to a over the counter Zopiclone and , the different experience in the US.

Please, address all issues raised and return a revised version for further analysis. 

We would appreciate receiving your revised manuscript by Dec 20 2019 11:59PM. To enhance the reproducibility of your results, we recommend that if applicable you deposit your laboratory protocols in protocols.io, where a protocol can be assigned its own identifier (DOI) such that it can be cited independently in the future. For instructions see: http://journals.plos.org/plosone/s/submission-guidelines#loc-laboratory-protocols

•A rebuttal letter that responds to each point raised by the academic editor and reviewer(s). This letter should be uploaded as separate file and labeled 'Response to Reviewers'.

•A marked-up copy of your manuscript that highlights changes made to the original version. This file should be uploaded as separate file and labeled 'Revised Manuscript with Track Changes'.

•An unmarked version of your revised paper without tracked changes. This file should be uploaded as separate file and labeled 'Manuscript'.

We look forward to receiving your revised manuscript.

 Kind regards,

 Cesario Bianchi

 Academic Editor

 PLOS ONE

 Journal Requirements:

Access to the French national health insurance database is subject to authorisation from the French National Health Data Institute (Institut National des Données de Santé; INDS, https://www.indsante.fr/) and from the French national commission governing the data privacy laws (Commission Nationale Informatique et Liberté; CNIL, https://www.cnil.fr/).

3. Thank you for including your ethics statement:

This study was approved by the local ethic board and the French National Data Protection and Freedoms Committee (number 918201). The study was registered on www.clinicaltrial.gov under the number NCT03584542. The study protocol is available as an open publication [8].

Additional Editor Comments:

 Dear Dr. Caillet,

Your manuscript was reviewed by 2 experts that found it interesting. However, they are asking for several clarifications and the inclusion of additional references.

In particular, both reviewers are concerned about the possibility that the decrease in Zolpedem Use be related to change to other drugs like diazepam (reviewers 1 and 2 and Zopiclone (reviewer 2) that is purcahsed over the counter in some European countries.

Is there an increase in its illegal use? Are there age-groups differences?, etc...

Many of these questions are still under investigation in the other studies encompassed in the whole ZORRO project. Regarding the specific question of the change in prevalence in zolpidem, by definition (claims data), the database that was used to treat the specific question of the change in prevalence in zolpidem use does not include information regarding OTC purchases and black market. However, we are currently conducting within the ZORRO project other studies specifically design to bring answers regarding switch (including to diazepam), variation of treatment’s episodes’ length, change in prescription habits from the physician and consumption habits from the patient that occurred after the change in law. Some of these researches use different methodologies (field studies), more suited to their respective objectives. The set of studies that are conducted are described in detail in our published protocol (Gérardin M, Rousselet M, Caillet P, Grall-Bronnec M, Loué P, Jolliet P, et al. French national health insurance database analysis and field study focusing on the impact of secure prescription pads on zolpidem consumption and sedative drug misuse: ZORRO study protocol. BMJ Open. 2019;9: e027443. doi:10.1136/bmjopen-2018-027443).

Reviewer 1 is also asking to make a comparison between US and France because the US experience seems to be different. The reviewer has many concerns about some terms use like addition vs dependence.

Both reviewers asked to include some references relevant to the topic. 

We added references and modified our manuscript accordingly.

Please, carefully address every issue raised and make changes if find necessary, in your revised version.

Thank you,

Cesario Bianchi

Reviewers' comments:

Reviewer's Responses to Questions

Comments to the Author

 1. Is the manuscript technically sound, and do the data support the conclusions?

Reviewer #1: Yes

Reviewer #2: Partly

2. Has the statistical analysis been performed appropriately and rigorously? 

 Reviewer #1: I Don't Know

 Reviewer #2: N/A

3. Have the authors made all data underlying the findings in their manuscript fully available?

Reviewer #1: Yes

Reviewer #2: Yes

4. Is the manuscript presented in an intelligible fashion and written in standard English?

Reviewer #1: Yes

Reviewer #2: Yes

5. Review Comments to the Author

Reviewer #1: Overall a nice paper demonstrating that a French government intervention had its intended effect of decreasing prescriptions of the drug zolpidem due to concerns about dependence. Contrasts with the US experience related to Drug Safety Communications and zolpidem prescribing.

Abstract

Lines 16-17: Not really a comment limited to the Abstract, but why only 6 months pre- and post-?

The first communication of ANSM about the measure took place in January 2017 and the application of the decree in April 2017. We therefore chose the beginning of the post-period when we thought that zolpidem consumption would have stabilized, i.e. in July 2017. In order to neutralize a potential seasonal effect, we sought to have the same monthly composition for each comparative period. So, each of the two comparative periods begins on July and ends on January (6-month period).

Introduction

Line 46: Would not use the term "addict patients," that term has fallen out of favor in the US and elsewhere. Suggest sticking to the term dependent as it was used earlier in the same sentence

We have amended the text accordingly.

Lines 48-49: Should read "increased their doses considerably" -- but please clarify what is meant by considerably. Do you mean 2.5 mg to 10 mg? If so I'm not sure that's a considerable increase since it's within the recommended dose range and patients are generally titrated up to achieve therapeutic effect

We have deleted “considerably” as it was confusing. In Victorri-Vigneau et al. [4], French reports of dependence to zolpidem reported an increase in zolpidem doses to high doses: mean ranging from 94mg to 265mg between 2004 and 2010.

Line 57: Please explain what "secure prescription pads" are. We don't have them in the United States, but this is an interesting concept and may have promise in the US depending on what they are and how they are utilized in France. Without a clear definition, it's hard to determine the broader applicability of these findings.

We have added an explanation on “secure prescription pads” in the Introduction section:

In light of these results, the prescription of zolpidem on secure prescription pads was put forward by the French National Commission on Narcotics and Psychotropic Substances. Secured prescription pads contains specific prescription sheets that are protected against forgery and that must be used by physicians when prescribing narcotics. On the 11th of January 2017, the ANSM decreed that as from April the 10th 2017, the prescription of zolpidem was to be done with such secured prescription pads in order to limit the risk of abuse and misuse. Moreover, such prescription must now display the number of therapeutic units, quantity and dosage written in full and for a maximum duration of 28 days.

Materials and Methods

Lines 80-82: A lot of acronyms in the Methods, some undefined (OSIAP, SNIIRAM). Can you write out the ones that don't appear as frequently in the manuscript (as you have with PMSI and CepiDC)?

We have written out the acronyms OSIAP, SNIIRAM and CNSP in full in the Introduction and the Materials and methods sections.

Lines 104-106: Date ranges don't make sense here. Please clarify. The second one goes (backwards?) from July 2017 to January 2017?

The second period is July 2017 to January 2018. We have made the change in the Materials and methods section.

Primary Outcome

Line 167: The use of the term "significant" here is confusing -- unless referring to "statistical significance" I would change this term to something else. I also think there is more to be presented here in terms of the sensitivity analysis, instead of just "no significant changes."

We agree with your point. We have changed the term in Results and added the results of sensitivity analysis as a supplementary table (S1 Table).

Lines 174-178: There may be a more concise way of saying all this. Both sentences seems to be pointing to the same data points, right? Decreasing from 183 to 79 is the 57% decrease? If that is not the case, more clarification is necessary.

Indeed, the two sentences repeated the same data points. We made the clarification in the Results section.

Discussion

Line 220: Would again avoid using the term "addiction". Dependence is more appropriate. Though I would be careful with use of the term dependence as well. How is it defined, and can your data really show that "dependence" among individual users decreased in this large, aggregate data set?

We agree with your comment. According to Diagnostic and Statistical Manual of Mental Disorders, 5th edition, dependence is usually defined using clinical criteria which are not available in medico-administrative data (e.g. tolerance, withdrawal). Consequently, we are not able to show a decrease of dependence to zolpidem clinically from our medico-administrative data. However, the decrease in the availability of the substance through the enforcement of the use of secured prescription sheets will possibly decrease the abuse and dependence to zolpidem. We will investigate this finding in the additional field study. We have changed the term “addiction” and moderated our phrasing in the Discussion section regarding to the lack of clinical data.

Lines 237-238: "may have successfully turned patient initially reluctant" (phrasing here doesn't read correctly and should be amended)

We have amended the phrasing to make it clearer.

Lines 251-252: "in a current context where the number of active practitioners in France is insufficient to meet the primary care needs." (this phrase, though perhaps true, is not relevant to this manuscript and should be deleted)

We have deleted this phrase.

Lines 253-254: This is an important distinction. It is notable that in the US, a Drug Safety Communication from the FDA did not result in statistically significant changes in prescribing (Kesselheim AS, Donneyong M, Dal Pan GJ, Zhou EH, Avorn J, Schneeweiss S, Seeger JD. Changes in prescribing and healthcare resource utilization after FDA Drug Safety Communications involving zolpidem-containing medications. Pharmacoepidemiol Drug Saf. 2017 Jun;26(6):712-721.) I would ask that the authors cite this paper and discuss the differences between France and the US that may have made secure prescription pads successful in France but DSCs unsuccessful in the US.

Secure prescription pads and Drug Safety Communications are completely different. In France, the use of secure prescription pads for some drugs is required by the law. It implies that a drug must be prescribed on a particular prescription support which is protected against forgery. This measure adds constraints on the prescription process. We have added the reference in our explanation on the difference between adding prescribing constraints and informing physicians about drug use. We have also explained secure prescription pads in the Introduction section.

Lines 259-260: Please also cite the US paper looking at news media and zolpidem prescribing changes: Woloshin S, Schwartz LM, Dejene S, Rausch P, Dal Pan GJ, Zhou EH, Kesselheim AS. Media Coverage of FDA Drug Safety Communications about Zolpidem: A Quantitative and Qualitative Analysis. J Health Commun. 2017 May;22(5):365-372.

We have added this paper in our references.

Lines 269-271: "An interesting point would be to monitor the use of long half-life benzodiazepines (e.g diazepam), which could give clues regarding frequency of tapering drug use after the measure" (I'm not sure what point is being made here; it should either be clarified further or deleted)

In this phrase, we wanted to explain that investigating the switches from zolpidem to other drugs after the measure is the key to understanding the decrease in use of zolpidem. Indeed, Touchard et al. showed an increase in prevalence of zopiclone reimbursements one year after the measure. This paper was published after our first submission to Plos One and has been added to references (number 10). Monitoring the use of alternative therapies (e.g. zopiclone, benzodiazepines…) after the measure will be the subject of a forthcoming publication. We have explained this point in more detail in the Discussion section.

Finally, I'd like the authors to take some time in the discussion to discuss the broader global implications of such a finding. Have secure prescription pads been used for other drugs in France? What was the outcome? Could this be replicated in a place like the United States, where health systems are far more fragmented? A succinct policy take-home would be nice. As the majority of cites are for French studies, a review of the US (and other EU) literature is warranted.

We have added a part in the Discussion to detail broader implications of the regulatory change and the previously use of prescription pads in France. We have also mentioned the potential to replication in other countries or other potential keys to reduce zolpidem use.

Conclusion

Lines 284-287: Sentence does not read coherently as written and needs to be rephrased.

We have rephrased it in a more comprehensive way.

Line 285: Would not use the phrase "abuser profiles"

We have changed the term “abuser profiles”.

Figure 1: Would use the month in the x-axis, not the date as shown. For example, July 2016, not 01/07/2016.

We have made the changes in the x-axis in Figure 1.

Figure 1: Please label the dashed lines more appropriately. I can't tell what 11/01/2017 and 10/04/2017 refer to here (and again, the US/EU conventions for dates leave me confused as to when these events happened.

We have made changes in labels of the dashed lines and changed the date format to make Figure 1 clearer.

Reviewer #2: This study aimed to analyse the effect of a change in French law that categorised Zolpidem to a higher level of prescription control in January 2017. Prior to the law change, Zolpidem use had been associated with tolerance and dependence mechanisms when administered in both a clinical and recreational setting. The law was changed to permit Zolpidem use only with Secure Prescription pads.

 Zolpidem is a non-benzodiazepine drug that binds to GABA-A receptors at the same site as traditional benzodiazepines. It acts as an allosteric modulator to potentiate the action of the natural agonist GABA and increase the channel opening frequency allowing chloride ions to enter the cell and hyperpolarise the membrane. The effects of this are varied including, sedation, anxiolysis, anaesthesia and anticonvulsant. Zolpidem being different structurally to benzodiazepines has preference for the alpha-1 subtype of the GABA-A receptor and thus causes a sedation effect without the other benzodiazepine associated effects. Interestingly Zolpidem has been associated in some cases with feelings of euphoria and hallucinations upon administration, an effect distinct from the GABA-A action, when co-administered with BZ, but when BZ was withdrawn, the euphoria persisted [1]. Perhaps Zolpidem may show secondary targets.

 This informatics study selects data from the General Sample of Beneficaries which is further connected to 3 other databases that banks patient information from over 98% of the French population. Participants of over 18 years of age were selected and data was analysed for ’prevalent users’ (users that had been reimbursed for zolpidem at least once before) and ‘incident users’ (users that has started afresh Zolpidem use). These parameters were compared with factors such as age, sex, health/disease state.

 The authors found that the change in law showed a negative correlation with Zolpidem use both in prevalent use and incident use.

 Criticisms:

 - The prevalent user data could not be analysed for separate age groups? Dependence in the elderly population may be higher due to reduced half-life of Zolpidem. There are many reports of Zolpidem toxicity in the elderly [1][2][3].

We agree with your comment. Dependence in the elderly is a major of concern. We included age groups in the multiple logistic regression (Table 3). Age was positively associated to zolpidem consumption with a very excess of risk of zolpidem consumption in the elderly population: OR = 8.86 [8.09-9.69] for ≥70y old vs 18-30y old. This finding highlights the need to further study the impact of the measure in the elderly population. A project is in currently in progress to study benzodiazepines and Z-drugs consumption in the elderly population including a focus on the evolution of zolpidem consumption after the measure. We have added a point about the elderly population in the Discussion section.

 - The authors mention themselves that the study is limited by the lack of information on how use has changed in correlation to use of alternative therapies, which may or may not be more dangerous (such as opioid or benzodiazepine use). Is it possible for the authors to access participant records to check for increase in prescriptions of these to find if there is correlation.

We agree with your comment. Studying the use of alternative therapies is the key to understanding the decrease of zolpidem use after the measure. In the French national health insurance database, we have information about reimbursed drug deliveries. So, we can study treatment transfers to other drugs (e.g. benzodiazepines, zopiclone…). Touchard et al. [10] recently published an increase in the prevalence of zopiclone reimbursements one year after the measure using the medico-administrative database. Monitoring the use of alternative therapies (e.g. zopiclone, benzodiazepines…) after the measure will be the subject of a forthcoming publication. We have made a clarification about this point in the Discussion section.

- The study is also limited in its lack of information regarding drug use/purchase (of both zolpidem or other therapies) outside of the awareness of the medical practitioners prescribing. Has illegal use/purchase of zolpidem increased in this time period?

We don’t have the information of the evolution of illegal use or purchase in the study period yet. These results will be available in our further national survey of forged prescription sheets OSIAP (Suspect Prescriptions Possibly Indicating Abuse or “Ordonnances Suspectes Indicateur d’Abus Possible” in French). Moreover, this will be also investigated in the field study. We have added this point in the Discussion section.

 - The authors could collect data on users over the period regarding what therapeutic strategies were used to reduce Zolpidem use, perhaps finding correlation with replacement therapies both pharmacological and psychological.

In the French National Health Insurance data, we cannot identify pharmacological and psychological therapeutic strategies to reduce zolpidem use. We agree that this is a very important point in order to understand fully the decrease in zolpidem use after the measure and this will investigated in our field study among problematic users and general practitioners. We have added a precision about this point in the Discussion section.

 - Zopiclone is an alternative to Zolpidem and in some European countries is available without prescription. How can we be certain that users are not switching to Zopiclone?

Zopiclone is indeed an alternative to zolpidem. Touchard et al. described an increase in prevalence of zopiclone one year after the measure. Thus, the switch from zolpidem to zopiclone may explain part of the observed decrease in zolpidem use. In a study that we are currently conducting, we planned to monitor the use of alternative therapies such as zopiclone or benzodiazepines in order to distinguish the switch to other drugs from the complete stop of hypnotic drug use. We have explained this in more detail in the Discussion section.

---

## [Decision Letter · Decision Letter 1]

17 Jan 2020

Prevalence of zolpidem use in France halved after secure prescription pads implementation in 2017: a SNDS database nested cohort study.

PONE-D-19-22052R1

Dear Dr. Caillet,

We are pleased to inform you that your manuscript has been judged scientifically suitable for publication and will be formally accepted for publication once it complies with all outstanding technical requirements.

With kind regards,

Cesario Bianchi

Academic Editor

PLOS ONE

Additional Editor Comments (optional):

Dear Dr. Caillet,

Thank you for carefully revising your manuscript that I now find acceptable for publication.

Please check the minor comments of reviewer 2, recommending further English editing.

thank you

Reviewers' comments:

Reviewer's Responses to Questions

**Comments to the Author**

1. If the authors have adequately addressed your comments raised in a previous round of review and you feel that this manuscript is now acceptable for publication, you may indicate that here to bypass the “Comments to the Author” section, enter your conflict of interest statement in the “Confidential to Editor” section, and submit your "Accept" recommendation.

Reviewer #2: All comments have been addressed

2. Is the manuscript technically sound, and do the data support the conclusions?

Reviewer #2: Yes

3. Has the statistical analysis been performed appropriately and rigorously? 

Reviewer #2: Yes

4. Have the authors made all data underlying the findings in their manuscript fully available?

Reviewer #2: Yes

5. Is the manuscript presented in an intelligible fashion and written in standard English?

Reviewer #2: No

6. Review Comments to the Author

Reviewer #2: Thank you for addressing our reviewing comments, we recommend that you check some grammatical errors in the manuscript before publication. Sometimes you have used patient in singular forms instead of the required plural form. Also limit should be referred to as limitations.

7. PLOS authors have the option to publish the peer review history of their article (what does this mean?). If published, this will include your full peer review and any attached files.

Reviewer #2: Yes: Gilberto DeNucci, Charles Serpellone Nash

---

## [Editor Report · Acceptance letter]

7 Feb 2020

PONE-D-19-22052R1 

Prevalence of zolpidem use in France halved after secure prescription pads implementation in 2017: a SNDS database nested cohort study. 

Dear Dr. Caillet:

I am pleased to inform you that your manuscript has been deemed suitable for publication in PLOS ONE. Congratulations! Your manuscript is now with our production department. 

With kind regards,

on behalf of

Dr. Cesario Bianchi 

Academic Editor

PLOS ONE